# Rapid Minimum Inhibitory Concentration (MIC) Analysis Using Lyophilized Reagent Beads in a Novel Multiphase, Single-Vessel Assay

**DOI:** 10.3390/antibiotics12111641

**Published:** 2023-11-19

**Authors:** Tejas Suresh Khire, Wei Gao, Brian Bales, Kuangwen Hsieh, Greg Grossmann, Dong Jin M. Park, Christine O’Keefe, Arnyah Brown-Countess, Sara Peterson, Fan-En Chen, Ralf Lenigk, Alex Trick, Tza-Huei Wang, Christopher Puleo

**Affiliations:** 1GE Global Research, Niskayuna, NY 12309, USAggrossmann@comcast.net (G.G.); peterson.sara.kelly@gmail.com (S.P.); lenigk@ge.com (R.L.); 2Department of Mechanical Engineering, Johns Hopkins University, Baltimore, MD 21218, USA; khsieh4@jhu.edu (K.H.); dpark55@jhu.edu (D.J.M.P.); cokeefe6@jhmi.edu (C.O.); thwang@jhu.edu (T.-H.W.); 3Department of Biomedical Engineering, Johns Hopkins University, Baltimore, MD 21218, USA; fchen24@jhmi.edu (F.-E.C.); atrick1@jhu.edu (A.T.)

**Keywords:** antimicrobial susceptibility testing (AST), antimicrobial resistance (AMR), lyophilized reagents, polymerase chain reaction (PCR), microfluidics, diagnostics

## Abstract

Antimicrobial resistance (AMR) is a global threat fueled by incorrect (and overuse) of antibiotic drugs, giving rise to the evolution of multi- and extreme drug-resistant bacterial strains. The longer time to antibiotic administration (TTA) associated with the gold standard bacterial culture method has been responsible for the empirical usage of antibiotics and is a key factor in the rise of AMR. While polymerase chain reaction (PCR) and other nucleic acid amplification methods are rapidly replacing traditional culture methods, their scope has been restricted mainly to detect genotypic determinants of resistance and provide little to no information on phenotypic susceptibility to antibiotics. The work presented here aims to provide phenotypic antimicrobial susceptibility testing (AST) information by pairing short growth periods (~3–4 h) with downstream PCR assays to ultimately predict minimum inhibitory concentration (MIC) values of antibiotic treatment. To further simplify the dual workflows of the AST and PCR assays, these reactions are carried out in a single-vessel format (PCR tube) using novel lyophilized reagent beads (LRBs), which store dried PCR reagents along with primers and enzymes, and antibiotic drugs separately. The two reactions are separated in space and time using a melting paraffin wax seal, thus eliminating the need to transfer reagents across different consumables and minimizing user interactions. Finally, these two-step single-vessel reactions are multiplexed by using a microfluidic manifold that allows simultaneous testing of an unknown bacterial sample against different antibiotics at varying concentrations. The LRBs used in the microfluidic system showed no interference with the bacterial growth and PCR assays and provided an innovative platform for rapid point-of-care diagnostics (POC-Dx).

## 1. Introduction

Antimicrobial/antibacterial resistance (AMR) is defined as the ability of the organism (bacteria) to withstand the bacteriostatic and bactericidal activity of the administered drug. The World Health Organization (WHO) considers AMR as the largest global threat to public health [1]. As per the 2019 antibacterial resistance report from the CDC, there are over 2.8 million antibiotic-resistant infections across the US resulting in more than 35,000 deaths [2]. The continuous evolution of bacterial species leading to the acquisition of AMR-determinant genes [3] and the reduced pace of the development of new antimicrobial drugs combined with the gross misuse of the available antibiotics are some of the main factors responsible for the rise of AMR [4]. While the empirical use of antibiotics is inevitable in emergency settings, the existing delay between blood culturing and the knowledge of its in vitro susceptibility data reduces the overall efficacy of this empirical treatment. In fact, in one of the largest retrospective cohort studies, it was found that one in five patients in the US with bloodstream infections received discordant empirical antibiotic therapy (i.e., a pathogen was treated with an antibiotic to which it was found not susceptible in vitro upon subsequent culturing) [5]. Studies like these highlight the importance of the availability of rapid (molecular) diagnostic tests in reducing the time to antibiotic administration (TTA) to identify resistant strains of infecting bacteria for reducing patient mortality and subsequently abating the crisis of increasing antibiotic resistance [6,7]. Thus, antimicrobial stewardship (optimized used of antibiotics) can be achieved by developing faster and more reliable methods of antimicrobial susceptibility testing (AST), which will be critical to limit the misuse of antibiotics in the clinical setting [8,9].

AST studies have historically relied on traditional gold standard methods, such as agar dilution, broth macro/microdilution, disk diffusion, or Epsilometer test (Etest), that offer high sensitivity and specificity to analyze antimicrobial susceptibility [10]. Unfortunately, these culture experiments are often time-consuming and require highly specialized means for obtaining, storing, and transporting the sample specimen, and their efficacy is greatly dependent on the culturing method. Advancements in the field of molecular biology have enabled the development of nucleic acid amplification test (NAAT) technologies that are rapidly replacing cell culture methods to diagnose the presence of mono- and polymicrobial infections in complex biological matrices. NAATs (particularly PCR) offer many advantages over the gold standard culture method; in particular, the sample collection is less invasive (wound swab samples can also be used) and their storage and transport conditions are also less impactful on the final analysis [11]. Another important advantage of the NAAT over cell culture methods is its ability to process nonviable cells that further increases its robustness [12]. Finally, the ability of NAATs to reliably provide sample-to-answers in a matter of a few hours (versus a few days) makes them an ideal candidate for point-of-care diagnostic (POC-Dx) applications [13].

Traditionally, NAATs have been implemented to detect genotypic AMR by investigating genetic mutations associated with AMR [14]. However, genotypic approaches are unable to predict phenotypic resistance reliably and are often susceptible to false negatives due to the constant evolution of AMR genes in infecting pathogens [15,16]. Genotypic approaches also cannot provide information on minimum inhibitory concentration (MIC), a quantitative metric of AST, exemplifying the need for phenotypic assays [17]. NAAT can also yield false positives by detecting commensal pathogens over their infectious counterparts, as is often the case with *N. gonorrhoeae* [18]. Furthermore, due to the increasing reliance of clinical practitioners on NAAT compared to culture-based AST, there is scarcity of susceptibility data available to physicians that further shifts the treatment regimen from evidence-based to empirical [16]. On the other hand, it is well recognized that antimicrobial dosages below the MICs can increase the rate of treatment failure and lead to selection for resistant bacteria [19]. Hence, there is a need for NAAT assays to be accompanied by AST studies for predicting MIC values in a real-time (or near real-time) format (i.e., to leverage the advantages of NAAT without losing the valuable information generated from culture-based AST) [20]. This simultaneous implementation of both AST and NAAT assays for a targeted antibiotic recommendation in a reduced time has been the primary motivation behind this work.

To reduce the potentially complex nature of workflows (i.e., number of steps required, consumables utilized, manual intervention per step, sample loss/damage during transfer and handling, etc.) required to execute these assays, both AST and subsequent NAAT assays should be ideally streamlined into a single-vessel format. However, NAAT and AST assays involve different chemistries and require different incubation conditions and temperatures. While AST requires the growth of bacteria at 37 °C in the presence of physiological buffers and pH, NAAT (typically PCR) is performed at higher temperatures with very specific pH and reagent compositions (salt, media, etc.) to support uninhibited nucleic acid amplification and detection [21]. Hence, there needs to be a spatial segregation of these chemistries for effective implementation of these assays in a single-vessel format for reliable MIC analysis.

Finally, for efficient and multiplex testing of several antibiotics at different concentrations, there is a need for microfluidic platforms equipped with a facile sample-to-device interface and precise fluid titration. For a simplified POC-Dx workflow, it is desirable to achieve this without the aid of multiple sterile consumables and the necessity of any mechanically actuating (pump/valve/lock) components. A single injection of bacterial sample followed with an equivolumetric distribution into different reaction chambers, each having a unique antibiotic at a specified concentration, will greatly simplify the operation of the AST/NAAT assays. The presented work aims to address these unmet needs by combining our expertise in microfluidic engineering and the development of lyophilized reagents in a unique format that allows for rapid estimation of MIC values.

### Multiphase Reagent Storage and Single-Step Multiplexed Assays

The challenges of performing both bacteria cell growth and PCR analysis in the same reaction vessel have been overcome using a novel combination of two technologies. First, lyophilized reagent mixtures for AST and PCR were prepared in a dry bead format, and the beads were then loaded into AST and PCR reaction chambers in the same PCR tube and separated using a paraffin wax seal (Figure 1). To achieve this, a preweighed wax pellet was first added in an empty PCR vial and melted by raising the temperature to 65 °C and then the PCR bead was added to molten wax. The lyophilized PCR bead was determined to be compatible with exposure to the liquid wax, which completely covered the bead and shielded it from external reagents. Once the PCR tube was cooled down to room temperature, the wax solidified, encapsulating the PCR bead and creating a physical partition for the addition of a second bead containing the antimicrobial drug. Furthermore, each of the PCR vials was associated with a microfluidic manifold that enables precise distribution of a common input sample evenly into eight different vials. Each PCR vial was preloaded with different antibiotic beads (i.e., different drug or different concentration of the same drug) enabling a multiplexed MIC analysis.

The manifold was designed and optimized to yield an equal output (~10 µL) at each of its ends, thus, uniformly distributing the bacterial load within the sample among different test conditions. The key innovation advancing the application of this microfluidic platform is the development and optimization of dried lyophilized reagent beads (LRBs) for potential long-term storage and stabilization of sensitive biological reagents (i.e., antibiotic drugs and PCR reaction mixtures). Stabilization in the dried state enables multiplexing and high-throughput reagent preparation at the manufacturer’s site, eliminating the need for precise liquid metering and distribution during use. Furthermore, having all reagents (PCR and antimicrobial drug) in a lyophilized state increases their room temperature stability, reducing the probability of hydrolytic degradation of the required DNA primers and PCR enzymes during storage. In addition to providing room temperature stabilization of the PCR enzymes in the dried state, the highly porous nature of these beads provides for rapid dissolution in aqueous solutions. This is achieved through use of lyophilization-stabilizing components, which remain compatible with both the growth step of bacterial cells and the amplification step of bacterial DNA. This feature of LRBs makes this approach ideally suited for use in a single-vessel reaction.

As shown in Figure 1, each reaction vessel consists of a standard PCR vial preloaded with two different beads each containing an antibiotic drug for AST assay (green) and/or PCR reagents (primers, enzymes, dNTPs, etc.) (violet). The PCR bead is embedded in the solid paraffin wax environment, which melts at 65 °C, and therefore remains solid (protecting the PCR bead) during incubation or culture phase (i.e., 37 °C) but melts and releases the NAAT reagents during PCR (i.e., 95 °C). This unique choice of wax melting temperature creates a hydrophobic barrier that helps in shielding the PCR bead from dissolution during the bacterial/sample incubation step of the AST assay. To begin the assay, the sample is added to the PCR tube (through the manifold) dissolving the unprotected drug LRB (green) and exposing bacterial cells to the released antibiotic. The highly porous nature of these beads allows for near-instantaneous dissolution and antibiotic release into the solution. Following the incubation of bacterial cells in the presence of the antibiotic for ~3 h, the temperature is increased to 98 °C to initiate the PCR reaction. The transition from 37 °C to 98 °C (denaturation temperature for PCR) melts the wax. The liquid wax being less dense than water displaces the aqueous bacterial sample and floats above it. The phase inversion of the cell sample to the bottom of the tube facilitates rapid dissolution of the PCR bead releasing the PCR master mix reagents (i.e., primers, deoxy-nucleotides, DNA polymerase, intercalating dye for PCR, etc.) into the solution.

Thus, the wax not only compartmentalizes the assays in space and time, but also acts as a sample mixing agent (during phase inversion of the sample liquid) and an evaporation seal preventing sample evaporation during hot cycles of PCR. The near-boiling temperatures of the initial PCR denaturation step causes bacterial cell lysis and subsequent DNA release into the solution. This step was adequate to lyse the bacteria tested herein. However, lytic enzymes (for instance, lysostaphin) could also be stored in lyophilized form in future tests to enable enzymatic as well as thermal lysis capability, but were not required for the bacteria tested [22]. The amplification of the released DNA was then quantified using qPCR directly from the same PCR tube used for bacterial cell growth. By using both control (i.e., no-drug) and drug-containing chambers within the manifold, AST and MIC values can also be obtained. To do this, Ct values were measured for each PCR tube across the manifold array. Pathogens susceptible to the antibiotic drug at a particular concentration underwent inhibited growth and demonstrated a high(er) Ct value (due to lower starting DNA concentrations), whereas pathogens resistant to the same concentration exhibited uninhibited growth, reporting low(er) Ct values. The range of antibiotic concentrations in which Ct values transition from high to low can be established as the MIC value(s) of that antibiotic for the pathogen. Herein, we compared these PCR-derived MIC values to those previously reported using the Clinical and Laboratory Standards Institute (CLSI) standards.

## 2. Materials and Methods

### 2.1. Lyophilized Reagent Beads (LRBs)

Hemo KlenTaq was purchased from New England Biolabs as a glycerol-free solution. Tetracycline hydrochloride was purchased from Sigma-Aldrich (St. Louis, MO, USA). Forward and reverse PCR primers were purchased from Integrated DNA Technologies (Coralville, IA, USA). EvaGreen and ROX normalizing dye were purchased from Biotium (Fremont, CA, USA). All other chemicals were purchased from Sigma-Aldrich (St. Louis, MO, USA).

#### 2.1.1. Preparation of *E. coli*/16S-Specific qPCR LRB Beads

Ten µL-sized LRBs were prepared using the previously described method [23] and contained 150 mM tricene, pH 8.75, 12.5 mM ammonium sulfate, 8.75 mM magnesium chloride, 0.8 mM dNTPs (0.8 mM each: dATP, dCTP, dGTP, and dTTP), 0.75 µM forward (5′-GAA GAG TTT GAT CAT GGC TCA GAT TG-3′) and 0.75 µM reverse (5′-TTA CTC ACC CGT CCG CCA C-3′) primer, Hemo KlenTaq enzyme (15,000 U), 10 mM KCl, 1 mM Tris-HCl at pH = 7.4, 10 µM EDTA, 10 µM dithiothreitol, 0.05% Tween 20, 0.05% IGEPAL CA-630 (octylphenoxypolyethoxyethanol), 25 nM Eva Green, and 1 µM ROX normalizing dye in addition to the lyophilization stabilizing components as described in [23]. The LRBs were flash-frozen with liquid nitrogen prior to lyophilization, and the LRBs were stored desiccated until use. Although a formal batch to batch analysis was not performed, 6 separate batches of LRBs were manufactured over a 24-month period and used to collect data in this manuscript. Each batch of LRBs was used within 30 days of its initial production.

#### 2.1.2. Preparation of Tetracycline LRB Beads

Ten µL-sized LRBs were prepared using the previously described method [23] and contained 1 mg/mL tetracycline hydrochloride (Sigma T3383-25G). The LRBs were stored desiccated until use. All PCR and antimicrobial LRBs were stored in low-humidity environments at room temperature prior to use and utilized from within days of manufacturing to one month following their initial production.

### 2.2. Antimicrobial Susceptibility Testing: Single-Phase Assay

The potency of lyophilized beads of tetracycline hydrochloride against *E. coli* was tested using cell growth indicator and PCR-based assays and compared with the results obtained from same assays performed in the presence of a standard aqueous solution of the antibiotic.

Bacteria Culture and Preparation: Bacteria were sampled from active exponential phase 10 mL cultures inoculated with a single-colony pick from a stock agar plate and grown in a shaking 37 °C incubator. Stock plates were prepared by overnight outgrowth at 37 °C of streaks from −80 °C glycerol bacterial stocks, used within 7 days of revival and stored at 5 °C between uses. Bacterial concentration was determined by using McFarland turbidometry, using the polynomial best-fit equation Y (bacterial concentration × 10^8^ CFU/mL) = 29.857 X^3^ − 9.0883 X^2^ + 10.35 X + 0.5238, where X is an A_600_ value within the range of 0.05–0.7, and corrections were made for dilution factors. Absorbance at 600 nm was measured on a Molecular Devices M5 luminometer. Cell counts determined this way were verified separately either by direct microscopic observation in cell-counting chambers, by dilution plating for CFU number, or by using quantitative PCR of diluted bacterial heat lysates compared to pure genomic DNA copy number standards.

AlamarBlue Assay: Minimum inhibitory concentration (MIC) value for tetracycline against *E. coli* was determined by using microtiter plate-based antibacterial assay incorporating alamarBlue as an indicator of cell growth [24]. Briefly, each well in a 96-well plate contained 100 µL mixture of 10 µL of 500 *E. coli* cells/µL (ATCC 25922) suspended in cation-adjusted Mueller–Hinton broth (CAMHB), 50 µL of 1:1 serial dilution of tetracycline in CAMHB, 10 µL of alamarBlue HS Cell Viability Reagent (Thermo Fisher (Waltham, MA, USA), A50100), and 30 µL growth medium of CAMHB. For a positive growth control, the antibiotic was replaced with CAMHB, and sterility (no growth) control included solutions without any cells. The cells were then grown in the presence of different concentrations of liquid tetracycline including controls for 16 h at 37 °C without any agitation. Alternately, the cells were grown in the presence of lyophilized beads consisting of different concentrations of tetracycline or a drug-free bead (positive control) under similar conditions as the liquid-only drug samples. The 96-well plate was then imaged for detecting color change in the wells to assess the bactericidal activity of tetracycline at different concentrations and determine the corresponding MIC value.

Titration of tetracycline in PCR Assay: A sample of 10 µL of *E. coli* culture containing 5000 cells was grown for 3 h at 37 °C in the presence of a serial dilution of liquid tetracycline or no drug as described above. After this growth phase, 1 µL aliquot of the culture was added to a PCR tube containing 10 µL of PCR mixture. The details of the PCR mixture and primer information were as described earlier. The Ct values obtained were analyzed to estimate the MIC values and compare them with the MIC values obtained from alamarBlue assay.

### 2.3. Antimicrobial Susceptibility Testing: Multiphase Assay

Multiphase assay for AST involved two 10 µL-sized LRBs in a single 0.1 mL PCR tube (Figure 1 with one consisting of tetracycline at the desired concentration and one consisting of the PCR mixture). First, the PCR bead was embedded in the wax at the bottom of the PCR tube by dropping the bead in the molten wax and letting it cool down to room temperature. The paraffin wax volume was optimized to be 10 µL and was measured using mass/density information from the vendor (Sigma-Aldrich (St. Louis, MO, USA), 411663-1KG). The required weight of the wax was punched from a molten sheet of solid wax and added to a PCR tube, which was heated slightly beyond the melting point of wax (65 °C) prior to the addition of PCR bead. After solidification, the drug bead (or no-drug control bead) was placed on top of the solid wax. A sample of 10 µL of *E. coli* culture containing 5000 cells was introduced inside the tube and incubated at 37 °C for 3 h in the absence of any agitation as described above. The same tube was then placed in the PCR instrument to initiate the qPCR. The corresponding Ct values were analyzed to estimate the MIC values and compare them with the MIC values obtained from single-phase assays (PCR and alamarBlue).

### 2.4. Microfluidic Manifold—Assembly and Operation

The microfluidic manifold was designed to allow for equal and precise distribution of input sample into 8 different PCR vials attached at the output. It was fabricated from acrylonitrile butadiene styrene (ABS)-like clear material (WaterShed XC11122) using 3D printing from Protolabs (Maple Plain, MN, USA) at a resolution of ~100 µm. The manifold design consisted of a female luer lock inlet compatible with standard syringes with male luer lock interface, and 8 radially splitting connecting channels used for distribution of sample from the inlet to outlet(s)/PCR vials. The terminal end of each channel had a round connector to create a tight fitting with the PCR vial and a vent hole. The schematic and assembled picture of this manifold are shown in Figure 2. Polyetheretherketone (PEEK) capillaries (McMaster-Carr, Elmhurst, IL, USA) with inner diameter of ~0.125 µm and length of 1 cm, were attached to outlet ports at the end of the internal 3D-printed channels. The PEEK tubing was secured inside the groove of the outlet port and permanently attached using plastic epoxy. After the assembly, the manifold was rinsed with isopropyl alcohol (IPA) and dried using nitrogen gun. Prior to each use with biological samples, the manifold was first rinsed with deionized water and dried using nitrogen gun. Then, the manifold was sterilized by exposing the top and bottom sides of the device to the UV light in a biosafety cabinet for at least 30 min on each side.

For sample distribution, 0.1 mL PCR vials were first attached to individual outlet ports and 84 µL sample was pipetted inside the luer lock port of the dried manifold (the target dispensed volume per PCR tube was 10 µL per vial for a total of 80 µL). An empty 1 mL syringe was used to displace the liquid from the luer port into channels and finally inside PCR vials. Syringe actuation was carried out slowly to avoid uneven fluid distribution into channels. Once the entire sample volume was completely actuated into the individual PCR vials, the syringe was removed, and then reattached to the manifold, and a second actuation (with an empty syringe) of air pressure from the syringe further displaced any residual fluid retained on channels of the manifold inside the PCR vials. After the liquid distribution was complete, the syringe was removed and the manifolds along with PCR vials were gently tapped to allow the sample liquid to collect at the bottom of vials. Finally, vials were removed carefully and processed for AST/PCR assay. After use, the manifold was rinsed with IPA and dried as described above for reuse.

## 3. Results and Discussion

### 3.1. Lyophilized Reagent Bead Manufacturability and Use Cases

The Lyophilized Reagent Bead (LRB) format for stabilizing PCR enzymes in the dried state at an ambient temperature involves freeze-drying all PCR components (buffer, required enzyme cofactors, dNTPs, primers, and polymerase enzyme) excluding the template DNA using previously reported methods [24,25]. In addition to providing ambient temperature stabilization to the PCR enzyme, LRB components do not routinely interfere with PCR reactions and do not need to be removed prior to initiating the PCR reaction. The freeze-drying process was developed to yield spherical beads, which rapidly dissolve upon the addition of water. The LRBs used in this study were designed for use with specific DNA targets and contained the required forward and reverse primers as well as reporting dyes (i.e., EvaGreen) such that only water and template DNA needed to be added prior to the beginning of thermocycling.

The LRBs were prepared by flash-freezing droplets of the reaction mixture (typically 10 µL) that contained the lyophilization and PCR components in liquid nitrogen to yield spherical frozen beads. The beads were then freeze-dried using a slow temperature ramp from −46 °C to ambient temperature to maintain the porosity of the lyophilized bead when buffer salts and PCR reaction components were included. Prior work using enzymes other than Taq DNA polymerase showed that the presence of glycerol more than 1.15 percent by weight in the solution (prior to lyophilization) and/or the presence of high concentrations of salts can adversely affect the lyophilization process. This effect is believed to be the result of a freezing-point depression, which leads to the frozen lyophilized bead melting before the lyophilization process is complete and leads to a nonporous collapsed bead which can be very slow to dissolve with water addition. For this reason, Hemo KlenTaq, the amplification enzyme, was selected for PCR amplification in the presence of cell growth media due to its ability to function in the presence of common inhibitors, and it was ordered in a glycerol-free solution. Typical beads formed after the lyophilization process, demonstrating a bright white and porose appearance and allowed direct loading into the wax-containing PCR vials for processing and use in PCR assays, as seen in Figure 3.

### 3.2. Assay–Reagent Compatibility

To test the utility of LRBs for the AST and PCR assays, it was necessary to confirm the compatibility of the ingredients used in the composition of the beads with the reagents used in both the assays. Similarly, to test the feasibility of performing both growth and PCR assays in a single vessel, it was crucial to verify the compatibility of the reagents used in one assay with the other and in the presence of the lyophilization mixture. The high salt concentrations (and other components) in standard 100% CAMH are known to interfere with PCR reactions; therefore, we tested both the bacteria doubling rate/growth and PCR efficiency at multiple growth broth concentrations. Likewise, the single-vessel AST + PCR assay involved two lyophilized beads, one with the drug and the other with the PCR mixture. Cell growth occurs in the presence of a single bead consisting of the target drug, and the PCR is carried out when both the beads are dissolved in the solution. This exposes the PCR enzyme to twice the concentration of the lyophilization mixture (2×). To better study these interassay effects, we performed a simple 2-factor design of experiments (DoEs) to understand the influence of different concentrations of cell media and lyophilization mixtures on the performance of PCR (Figure 4A). The multiphase reactions were carried out in the presence of 60% or 100% CAMHB and 1× or 2× LRB mixtures (corresponding to 1 or 2 beads, respectively), with the center point at 80% CAMHB and 1.5× LRB mixture. The bacteria inoculum was fixed at 5000 cells. The output metrics analyzed included the PCR Ct values, PCR Tm (melting point), and delta Tm (i.e., difference between the melting of target and nonspecific amplicons). These experiments were performed in the absence of any drug or wax to simplify the initial DoE. The results are as described in Figure 4B–D.

Figure 4 shows important observations on the effects of higher salt concentrations (at 100% CAMH) and higher concentrations of polysaccharides from lyophilization mixtures on the overall performance of the PCR assay. Use of 100% CAMH resulted in increased variability in Ct values when the 1× LRB mixture was used. However, this variability was reduced when the 2× LRB mixture was used (as shown in Figure 4B). Likewise, the use of the 2× mixture reduced the variability in melting point (Tm) of the amplicons in the presence of 100% CAMH media (Figure 4C). The difference between melting points of specific and nonspecific amplification products was also more uniform in the presence of the 2× mixture as observed in Figure 4D. Tm analysis showed that the ratio of peak values of the negative first derivative (-∆F/∆T) of melting curves of specific to nonspecific amplicons was consistently higher than 1.7 across all the conditions tested. In conclusion, the presence of two beads in the reaction vessel (i.e., 2× mixture) alleviated variability introduced by 100% CAMH in the PCR reaction. These interaction effects highlighted that the presence of higher concentrations of the lyophilization mixture may improve the variability introduced by the higher concentration of salts present in the 100% CAMH mixture. It should be noted that nonreducing sugars typically utilized in lyophilization processes (e.g., trehalose and sucrose) are known to provide thermal denaturation protection to PCR (and other) enzymes [26]. These DoE results provide initial validation of the feasibility of a single-vessel format for both growth and PCR assays by establishing the compatibility within the different assay reagents.

### 3.3. MIC-AST Assays

After the initial validation of the performance of different reaction components in the single-vessel format (additional information in Appendix A), we next decided to study the effects of tetracycline on the growth of *E. coli* under different test conditions. Tetracycline is a short-acting antibiotic that inhibits bacterial growth by inhibiting protein translation. It binds to the 30S ribosomal subunit and prevents the amino acyl tRNA from binding to the A site of the ribosome [27]. Furthermore, the CLSI guidelines have established the MIC value for tetracycline against *E. coli* strain 25,922 in the range of 0.5–2.0 µg/mL [28]. This allowed for a comparison with published values and a validation of our novel multiphase MIC-AST assays. AlamarBlue^®^ (ThermoFisher, Waltham, MA, USA) is a commercially available fluorometric kit consisting of the oxidizing agent resazurin that changes color from blue to red in the presence of viable cells. The viable cells reduce the resazurin agent, and the corresponding color change is a quantitative metric that can be analyzed to estimate the viability and growth of cells. A total of 500 bacteria were grown in the presence of resazurin and increasing concentrations of tetracycline ranging from 0.0625 to 8 µg/mL. After 16 h of growth at 37 °C, there was no observable color change for drug concentrations higher than 0.5 µg/mL, which is in agreement with the CLSI’s established range of 0.5–2 µg/mL (Appendix A). We then repeated this assay but using the lyophilized version of tetracycline instead of its liquid form. The absence of color change was observed for concentrations higher than 0.5 µg/mL, which confirms that the MIC value for lyophilized beads matches that using the standard liquid antibiotic solution (Appendix A). This not only validated that the lyophilization process preserves the efficacy of the dried antibiotic, but also demonstrated that the LRB mixture does not interfere with cell growth or the bactericidal activity of the antibiotic.

After initial validation with cell growth experiments, we proceeded to test the feasibility of using the PCR assay for estimating the MIC values in our multiphase reaction in the single-vessel design. First, we embedded PCR LRB in the wax at the bottom of the PCR vial to protect the PCR bead from external liquid manipulation. Next, we placed the antibiotic LRB on top of the solid wax column. The addition of the bacterial sample immediately dissolved the bead releasing the antibiotic in the solution and exposing the bacterial cells to the drug, without affecting the wax or the embedded PCR bead. Cell growth in the presence of varying concentrations of tetracycline for 3 h was sufficient for the antibiotic to inhibit cell multiplication due to its rapid action. After 3 h of incubation at 37 °C, PCR was performed to assess the inhibitory effects of tetracycline on bacterial growth. The threshold value (Ct) of the PCR reaction is an objective metric to compare the initial genomic load between different reactions: higher Ct values imply lower template DNA concentrations and vice versa. To have a better comparison of the performance of the drug to the no-drug controls, the data were normalized by subtracting the mean Ct value for the no-drug (positive) control from individual Ct values. The efficacy of antibiotic LRB was verified by running the same assay in parallel but replacing the antibiotic beads with liquid tetracycline at equivalent concentrations to test for any effects of lyophilization on the antibiotic performance. The comparison of these two studies is summarized in Figure 5.

The MIC value of an antibiotic is the lowest concentration beyond which no bacterial growth is observed. As such, the delta Ct value should stop increasing for the reaction(s) run in the presence of drugs with antibiotics at concentrations higher than the MIC value. The plateauing of (delta) Ct values in Figure 5 can be observed at concentrations > 2 µg/mL—for both liquid and dried antibiotics—confirming the CLSI’s established MIC standards (0.5–2 µg/mL). In certain cases where the PCR bead or the antibiotic bead was visibly observed to be incompletely dissolved after a PCR run, the datapoint was excluded from any analysis. This result establishes that PCR data can be used for determining the MIC value of an antibiotic drug and it also demonstrates that lyophilization of PCR reagents does not affect or alter their chemical reactivity.

### 3.4. Microfluidic Assay for MIC Estimation

To increase the throughput of the sample analysis for AST/MIC, we designed a precise microfluidic manifold that can perform multiple AST assays using a single input of a bacteria-containing sample. This multiplexing is essential in a clinical environment where a multitude of samples or assay conditions need to be processed within a short timeframe. The microfluidic manifold consists of a single inlet and eight outlets but can be scaled to larger formats by increasing the fluidic splitting in the manifold. The manifold channels and their dimensions were carefully optimized to yield an equal volume distribution within a resolution of ~10 ± 1 µL (measured using water weight. Prior to each test, the manifolds were used to dispense cell media without any spiked bacteria and the collected samples were analyzed by using PCR to serve as a sterility verification (i.e., no visible amplification). For every subsequent manifold run, the attached PCR tubes consisted of both beads for the AST and PCR assays, respectively. The antibiotic beads were of different concentrations and were also present in a drug-free format to serve as an internal positive growth control. Thus, every manifold had drug LRB spanning the entire concentration range to enable standalone testing of MIC across multiple drug conditions. The normalized PCR-Ct values for all vials at the same drug concentration within four separate manifolds are plotted in Figure 6. The delta Ct or the difference in the Ct values for the sample with drug and sample without drug (i.e., drug-free bead) was analyzed and compared to equivalent assay conditions prepared using manually pipetted samples (i.e., nonmanifold PCR tubes). The summary data are as shown in Figure 6.

Both data sets (i.e., the manifold-loaded and manually loaded samples) demonstrated growth inhibition at the same concentrations of the drug. That is, ANOVA tests confirmed that the data points for 2 µg/mL and 10 µg/mL for both methods demonstrated the absence of cell growth (i.e., were not statistically different). This validates the utility of the microfluidic manifold for further multiplexing antimicrobial susceptibility testing using the single-vessel format, without affecting the observed MIC values.

## 4. Conclusions

Herein, we demonstrate a novel two-phase AST/MIC analysis platform that uses two premade lyophilized reagent beads (LRBs) to fully automate the antimicrobial testing process. The unique format encases the PCR reagent bead within a sealed wax environment (prior to PCR), enabling bacteria culture and PCR testing in the same/single tube. Loading of the sample through a microfluidic manifold, enables single-step priming of all drug concentrations necessary for MIC testing. Use of the wax-sealed compartment also enabled protection of the PCR reagents/LRB during the culture/incubation phase. Post-incubation heating of the reaction vessel melts the wax and allows rehydration of the PCR reagents for subsequent PCR amplification and assessment with bacteria-specific PCR primers. Different levels of growth inhibition across the array of tubes (due to the presence or absence of different levels of the drug) can then be assessed based on CT values for the PCR reactions run across the array of tubes. As shown, this unique single-vessel/two-phase system enabled MIC testing with only a single-user step, which is in sharp contrast to the multiple sample preparation and pipetting steps necessary to run current gold standard tests, such as microbroth dilution or agar plate-based AST/MIC assessment.

While our results demonstrate proof of concept for using LRBs in a sealed wax environment for rapid/single-tube microbial detection and testing, additional work is needed for translation into clinical and field use. For example, one primary advantage of the standard CLSI methods of AST/MIC remain the universal nature of their application across clinical sample types (i.e., post-culture and microbial isolation), microbe species/strains, and antimicrobials. While universal/broad-spectrum testing is beyond the scope of this proof of concept demonstration and report, we show in Appendix A that in addition to tetracycline (shown above), ciprofloxacin can also be manufactured in lyophilized/LRB form and utilized for the PCR-based detection of antimicrobial growth inhibition. Furthermore, in Appendix A, MIC values are determined for *E. coli* within multiple concentrations of tetracycline (Tet), penicillin G (PenG), and ciprofloxacin (cipro) mixed/diluted directly in LRB reagents, thus showing overall compatibility with multiple antimicrobial types. We also tested these additional antimicrobials in the full wax-sealed, two-phase system and show that PCR-based growth inhibition is apparent using both PenG (Appendix A) and Cipro (Appendix A).

The LRB PCR formulation utilized herein is available commercially in several standard PCR mixtures and has already been used internationally across medical applications/fields [25]. However, we additionally show in Appendix A that our specific formulation (using either the Hemo KlenTaq (manuscript figures) or SSO Advanced (Bio-Rad; Appendix A) enzymes) is compatible with the amplification of multiple types of genomic DNA and organisms (i.e., *E. coli* and a. Baumannii). It is anticipated that the heat lysis-based method utilized herein may not be sufficient for some Gram-positive bacteria; however, our group [29] and others [30] have shown that additional enzymatic or mechanical lysis steps are available that are compatible with rapid bacteria identification and AST/MIC analysis. In addition, while we demonstrate in the manuscript that the two-phase, single-step PCR system is compatible with high salt environments (i.e., input sample contains CAMH growth buffer), we further show in the Appendix A that the PCR LRB is capable of amplification in other buffers that are commonly used as transport media for clinical swab samples. Finally, we show in Appendix A that the variation in sample/fluid splitting using the microfluidic manifold can provide a low coefficient of variation, and is capable of providing single-step fluid manipulations for AST/MIC assays. This additional data further show utility and readiness for further testing and application in clinical trials and translational activities. In addition, our group and others have demonstrated that simple sample preparation systems may further enable rapid AST/MIC testing from more complex clinical matrices, such as blood [29,30,31,32], and that rapid PCR assessment may be used to determine the lack of growth inhibition in samples containing drug-resistant organisms [9,20,29,31,32,33,34,35,36,37,38,39,40].

It should also be noted that while several rapid PCR-based phenomolecular AST/MIC assays are currently under development [9,20,29,31,32,35,38,39,40], their utility may ultimately depend on the development of standardized methods that consider specific microbe–antimicrobial biophysical interactions. As an example, we demonstrate in Appendix A that within the 3 h incubation period utilized herein, PenG demonstrates a nearly complete growth inhibition on *E. coli* (i.e., Ct values for the drug-treated samples are equal to gDNA controls with equal initial inputs), while ciprofloxacin demonstrates a partial inhibition (i.e., Ct values are less than the no-drug control, but not equivalent to the gDNA standard control). This may be due to the differences in the mechanism of action between PenG and Cipro on *E. coli*. PenG is known to inhibit cell wall biosynthesis, weaken wall integrity, and result in cell death due to a lack of maintenance of osmotic gradients. Cipro inhibits DNA gyrase and topisomerases, thereby inhibiting cell division in the near term, while DNA fragmentation will eventually result in cell death as a secondary/longer term effect (i.e, at later timepoints). Thus, as with CLSI’s standard methods, new standards must be developed that take into account different timing of the drug effect in these shorter incubation periods and PCR assays time windows.

Due to the increasing prevalence of resistant organisms and drug-resistant infections, the development of rapid AST/MIC technologies remains important research and a clinical goal [30,31]. However, in the last decade, a significant number of new techniques have been developed [9,30,31,32,33,34,35,36,37], including molecular-, genome-, and novel biophysical-based methods, to rapidly detect, identify, and characterize drug resistance in clinical, environmental, and food/pharmaceutical/industrial samples. Currently, the lack of adoption of these new technologies is in part due to (1) the cost associated with performing full clinical trials to fully validate new methods; (2) the difficulty displacing previously installed and systematically entrenched methods in large clinical (or other) environments; and (3) differences in the cost per assay for new technologies versus the relative cheap gold standard assay formats [30]. While their combination is novel, the lyophilized reagents and wax components themselves utilized herein are inexpensive and use of the multiphase reaction eliminates the need for any complex fluid handling required for traditional AST/MIC methods. While beyond the scope of this initial report, the compatibility with low-cost reagents and the simplicity of the single-vessel assay may facilitate further investment and clinical testing, which remains a goal of our team.

## Figures and Tables

**Figure 1 antibiotics-12-01641-f001:**
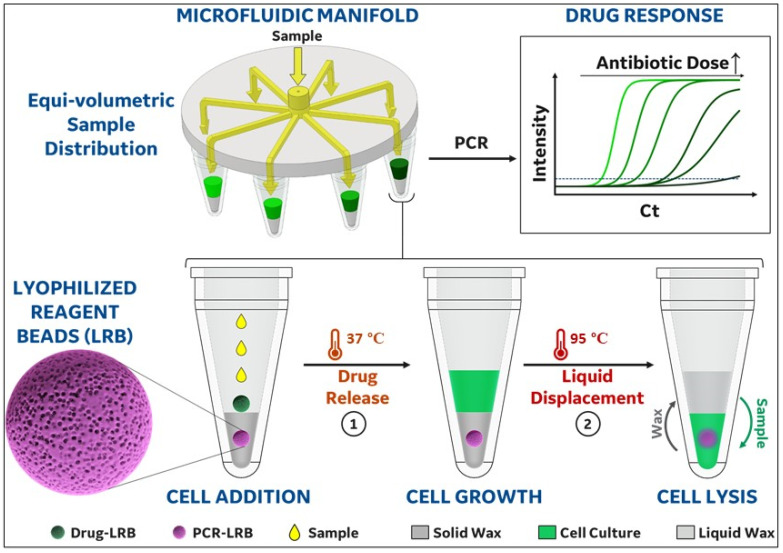
Eight PCR vials are attached to a microfluidic manifold that evenly distributes input sample into each vial. Each of these vials has different antibiotics or the same antibiotic at varying concentrations (as indicated by progressively darkening shades of green in both the manifold and drug response chart) with increasing concentrations. Inhibition of growth and bacteria doubling by the different concentrations of antibiotic results in the presence of different concentrations of bacteria following a brief culture period. The vials with less bacteria (due to growth inhibition) result in higher PCR Ct (cycle threshold) values after subsequent PCR. To achieve PCR readout, each vial consists of two lyophilized reagent beads (LRBs) containing (1) antibiotic drug at specified concentration (green bead) and (2) PCR reaction mixture including primer for amplification (violet bead). The PCR LRB is embedded in solid wax (dark gray, melting point = 65 °C), while the drug LRB is stored on top of the wax. The addition of sample (swab eluate/ isolated bacteria) dissolves the drug LRB immediately and releases the antibiotic drug in the growth medium added with the sample. Following the incubation of this mixture at 37 °C for the AST assay, the temperature is raised to initiate PCR reaction at 95 °C that melts the wax layer. The phase change of wax from solid to liquid reduces its density and hence allows it to displace and float on top (light gray) of the aqueous bacterial culture medium. The contact of this sample to PCR LRB facilitates the dissolution of PCR reaction mixtures and subsequently promotes amplification of the DNA released from bacterial cells, which undergo autolysis at ~95 °C. The times required for each step in Figure 1 will vary based on application and assay goals; however, herein, cell addition and drug release required seconds (i.e., only the time required for dissolution of the drug LRB); cell growth was performed for 3 h; wax melting and sample/PCR LRB inversion took place during the initial 95 °C initiation phase of the PCR reaction (minutes); and PCR cycling for this specific primer/PCR reaction was performed in under one hour.

**Figure 2 antibiotics-12-01641-f002:**
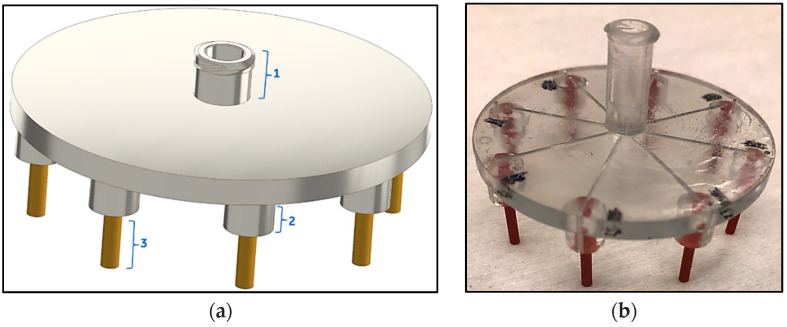
Microfluidic manifold schematic (**a**) and image (**b**). The manifold has 3 main elements: (1) female luer lock interface, (2) connector(s) to attach to PCR vial(s), and (3) PEEK tubing outlets for dispensing the liquid from the manifold to the inside of the PCR vial.

**Figure 3 antibiotics-12-01641-f003:**
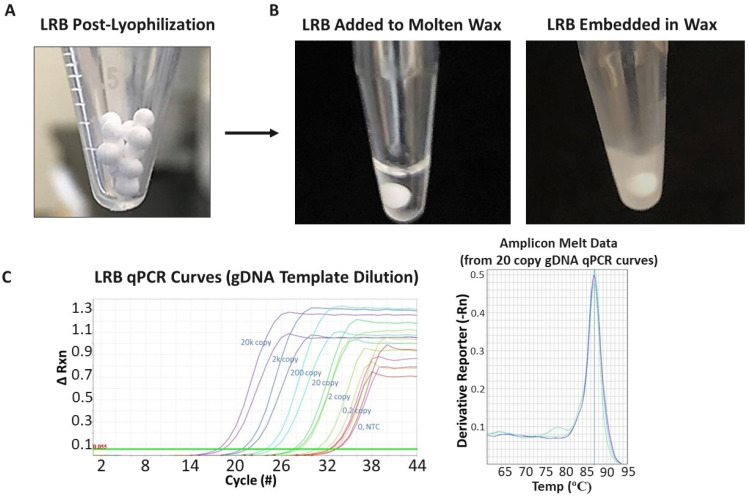
(**A**) Image of typical LRB bead appearance after initial manufacturing stage. Successful bead manufacturing produces drug and/or PCR lyophilized reagent beads (LRBs) that are bright white and porous in appearance, yet are solid enough for further manipulation (i.e., moving to individual PCR vials and processing/covering in wax). (**B**) For the multiphase reaction chambers, the LRBs are added to a chamber containing molten wax (i.e., maintaining the wax-containing tube above the wax melting temperature; left). Once the LRB is added, the temperature is lowered, the wax rehardens, and the LRB is then embedded and protected within the wax phase. (**C**) (**left**) qPCR curves from initial runs in which a sample containing increasing concentrations of prepurified genomic *E. coli* DNA was added to the reaction chambers (over the wax); the wax was then melted; the LRB mixed with the added sample; and PCR performed. qPCR response was observed (i.e., decreased cycle number in which fluorescence increased over background; left) to the increased gDNA template concentrations (purple, 20,000 copies; black, 2000 copies; blue, 200 copies; green, 20 copies; light green, 2 copies; yellow, 0.2 copies; and red, 0 copies/no template control (NTC)). The melt curves (**right**) from the 20 gDNA copy qPCR curves show a clean/single product PCR result with a melting point (86.93 degrees Celsius) that matches the expectation for the *E. coli* primer set. Each colored line represents the melt curve from one PCR tube/reaction.

**Figure 4 antibiotics-12-01641-f004:**
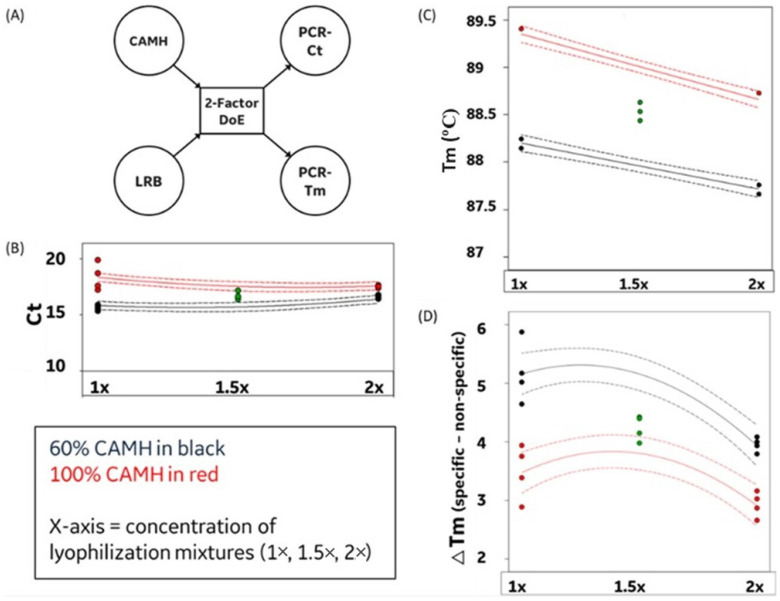
Scheme of the 2-factor Design of Experiment (DoE) (**A**). Interaction plots demonstrating the effects of increasing concentrations of lyophilization mixtures on the Ct value (**B**), melting point of target amplicon (**C**), and difference between the melting points of target and nonspecific amplicons (**D**). Curves in red represent 100% CAMH (cation-adjusted Muller-Hinton), and curves in black denote 60% CAMH. Green dots represent center point values (80% CAMH and 1.5× lyophilized reagent bead (LRB) mixture).

**Figure 5 antibiotics-12-01641-f005:**
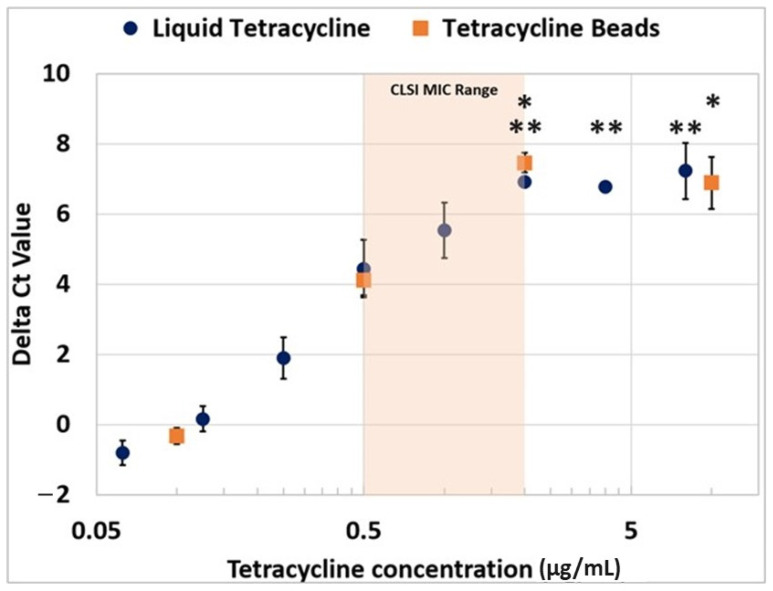
Comparison of delta Ct values obtained during PCR assay after growing *E. coli* in the presence of different concentrations of tetracycline hydrochloride in its liquid and lyophilized forms. The shaded region indicates the MIC values as established by CLSI guidelines. Error bars indicate standard deviation. A homoscedastic ANOVA test was performed using Tukey pairwise comparisons. Points in plots labeled with * (orange curve) or ** (blue curve) had the same Tukey pairwise groupings, denoting lack of a statistical difference of those points within the curve. Asterisks are provided for each data point on the plot (i.e., top asterisk is for the top data point).

**Figure 6 antibiotics-12-01641-f006:**
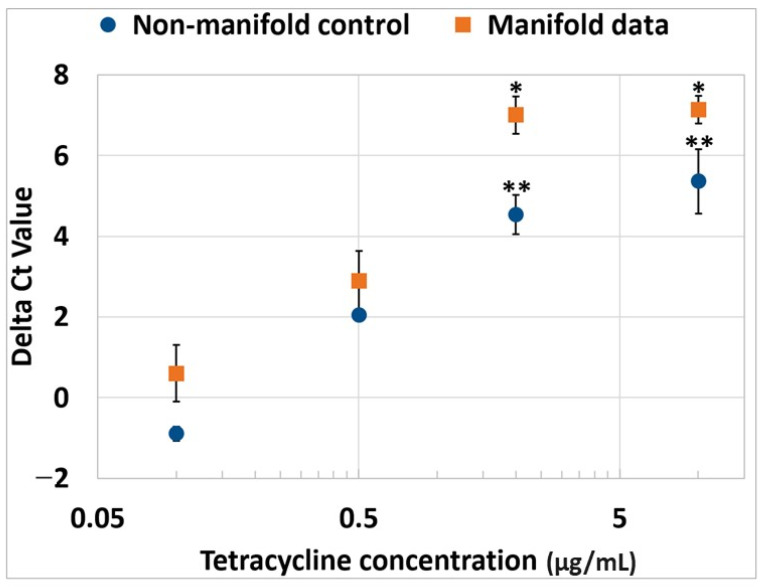
Comparison of delta Ct values from manifold and manual (standard PCR tube) controls. The Ct values obtained in the presence of control (no antibiotic) bead was subtracted from the Ct values obtained in the presence of an antibiotic bead to calculate the difference (delta Ct). Error bars indicate standard deviation. A homoscedastic ANOVA test was performed using Tukey pairwise comparisons. Points in plots labeled with * (orange curve) or ** (blue curve) had the same Tukey pairwise groupings, denoting lack of a statistical difference of those points within the curve.

## Data Availability

Data are contained within the article and Appendix A.

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
