# Peer review of "Rapid Minimum Inhibitory Concentration (MIC) Analysis Using Lyophilized Reagent Beads in a Novel Multiphase, Single-Vessel Assay"

_antibiotics, 2023, doi:10.3390/antibiotics12111641_

Round 1
Reviewer 1 Report
Comments and Suggestions for Authors
Authors report an AST/MIC platform that employed lyophilized reagent beads and a sealed wax environment to perform PCR on E coli cultures incubated with tetracycline of varying concentrations. Results agreed with standard AST. The use of wax materials stored in the PCR tubes was an innovative way to set up the assay.
The manuscript as it stands, appears to be in the proof-of-concept stage. Only 1 type of bacteria/1 antibiotic/1 gene were tested, which doesn’t demonstrate broad applicability of the technology. It is recommended that the authors try at least another type of bacteria/gene (something Gram-positive, like S. aureus, might be a good candidate to contrast E. coli), another antibiotic, and a strain that is resistant to the antibiotic being tested. It’s also unclear how the technology would be employed practically in its current set up. Is the intended use with clinical samples (if so, what type of sample?, what’s the expected concentration of bacteria in those samples?, and spiked or clinical samples should be tested)? Is the intended use with an overnight culture diluted to the desired concentration (in which case that should be considered in the overall assay time)?
Additional comments for each section are included below:
Introduction:
The introduction does a great job of discussing relevant literature, identifying advancements and limitations in the field.
Page 4- Figure 1. Including the timing of the incubations would be helpful to give readers a sense of the timing of each step.
Page 4- Line 151. It would be helpful to get a sense of how long the lyophilized materials are stable for (and under what storage conditions). See comment below in Results/discussion section.
Page 5- Line 186. Are there differences in lysis for Gram-positive and -negative bacteria? It is suspected that Gram-positive may be harder to lyse.
Materials and Methods:
Page 5- Line 210. Please specify which of the materials are the lyophilization-stabilizing components that are mentioned on page 5 line 161.
Page 6- Sections 2.2/2.3. How were the E. coli samples prepared? From an overnight culture, from a freezer aliquot? Were concentrations determined via OD600, followed by dilution?
Results and Discussion:
Page 13- Figure 6. In the caption it should be specified that the p values are from an ANOVA test (including the type of ANOVA done).
Page 13- Line 475. Specify the type of ANOVA test performed.
Data demonstrating the reproducible and uniform production of the beads should be included (in particular for the drug-LRB).
Data demonstrating the longevity of storage of the lyophilized materials (and under which conditions) should be included.
Conclusion:
The conclusion should provide more specific future directions of this work (in contrast to the current state of the work).
Author Response
We have included the revised manuscript with responses to reviewers' comments attached.

Reviewer 2 Report
Comments and Suggestions for Authors
In this paper, the authors paired the short growth period (~3-4 hours) with downstream PCR detection by microfluidic technology to provide phenotypic antimicrobial susceptibility test (AST) information, and finally successfully predicted the minimum inhibitory concentration (MIC) value of antibiotic therapy. It’s expected this could be an impactful paper after these improvements.
1. In the antibacterial experiment, bacterial concentration and drug concentration are two important factors. How did the author ensure the uniform distribution of bacterial quantity in the sampling process? Did the experimental results be controlled?
2. There are differences in the growth rate of different microorganisms, so how to find the most appropriate time in the process of verifying the action of drugs
3. There will be a sequence of substances in the freezing and thawing process of freeze-dried beads. How to control the influence on the experimental process
4. As an analytical method, authors should provide differences between batch parallelism experimental results
5. In Fig 5, the difference analysis between data should be added
6. Some figures in the figure have non-standard phenomena. For example, the unit of Fig 3c temp should be ℃ Fig4c melting point Tm should be added to the unit ℃
Author Response

(The authors gave the same response as above.)
